# Targeted Immuno-Antiretroviral to Promote Dual Protection against HIV: A Proof-of-Concept Study

**DOI:** 10.3390/nano12111942

**Published:** 2022-06-06

**Authors:** Subhra Mandal, Shawnalyn W. Sunagawa, Pavan Kumar Prathipati, Michael Belshan, Annemarie Shibata, Christopher J. Destache

**Affiliations:** 1School of Pharmacy & Health Professions, Creighton University, Omaha, NE 68178, USA; shawnalynsunagawa@creighton.edu (S.W.S.); pavankumarprathipati@creighton.edu (P.K.P.); chrisdestache@creighton.edu (C.J.D.); 2Department of Medical Microbiology & Immunology, Creighton University School of Medicine, Creighton University, Omaha, NE 68178, USA; michaelbelshan@creighton.edu; 3Department of Biology, College of Arts and Sciences, Creighton University, Omaha, NE 68178, USA; annemarieshibata@creighton.edu; 4Division of Infectious Diseases, School of Medicine, Creighton University, Omaha, NE 68178, USA

**Keywords:** dolutegravir, tenofovir, CCR5 monoclonal antibody, nanoformulation, targeting, functional HIV cure

## Abstract

The C–C motif chemokine receptor-5 (CCR5) expression on the T-cell surface is the prime barrier to HIV/AIDS eradication, as it promotes both active human immunodeficiency virus (HIV)-infection and latency; however, antiretrovirals (ARVs) suppress plasma viral loads to non-detectable levels. Keeping this in mind, we strategically designed a targeted ARVs-loaded nanoformulation that targets CCR5 expressing T-cells (e.g., CD4+ cells). Conceptually, CCR5-blocking and targeted ARV delivery would be a dual protection strategy to prevent HIV infection. For targeting CCR5+ T-cells, the nanoformulation was surface conjugated with anti-CCR5 monoclonal antibodies (CCR5 mAb) and loaded with dolutegravir+tenofovir alafenamide (D+T) ARVs to block HIV replication. The result demonstrated that the targeted-ARV nanoparticle’s multimeric CCR5 binding property improved its antigen-binding affinity, prolonged receptor binding, and ARV intracellular retention. Further, nanoformulation demonstrated high binding affinity to CCR5 expressing CD4+ cells, monocytes, and other CCR5+ T-cells. Finally, the short-term pre-exposure prophylaxis study demonstrated that prolonged CCR5 blockage and ARV presence further induced a “protective immune phenotype” with a boosted T-helper (Th), temporary memory (TM), and effector (E) sub-population. The proof-of-concept study that the targeted-ARV nanoformulation dual-action mechanism could provide a multifactorial solution toward achieving HIV “functional cure.”

## 1. Introduction

Antiretroviral therapy (ART) is the only treatment strategy for human immunodeficiency virus (HIV) infection. ART improves life expectancy by effectively controlling plasma viral load (pVL); however, it is unable to eradicate the virus. Patients have to commit to continual life-long ART. Additionally, ART stoppage can reactivate the latent virus; therefore, alternative ways are under investigation to search for potential candidates to ‘functional-cure’ HIV infection [1].

One of the essential targets of HIV research is the C–C motif chemokine receptor 5 (CCR5), a predominant co-receptor expressed on lymphocytes (such as CD4+ T-cells, latently infected cells, dendritic cells (DCs), and macrophages), responsible for HIV-1 cellular entrance [2]. Maraviroc, a CCR5 antagonist that blocks HIV entry by docking on the CCR5 receptor on CCR5+ cells, is the first approved CCR5-antagonist for HIV treatment [3]. Other promising approaches include genome editing that disrupts CCR5 alleles in CD4+ T-cells by infusing an engineered zinc finger nuclease (ZFN) [4]. Several other alternatives such as ribozymes [5], transcription activator-like effector nuclease (TALENs) [6], short hairpin RNAs [7], and the clustered regularly interspaced short palindromic repeat-Cas 9 (CRISPR/Cas9) nuclease system [8], has been investigated to promote CCR5-targeted gene-editing to ensure HIV protection; however, the only gene-therapy approach to date that has conferred HIV cure is the use of CCR5 delta32 natural mutant genotype. So far, HIV patients, the “Berlin patient,” “London patient,” the “Düsseldorf patient,” and the first woman, the “US Patient,” have reportedly been cured only after transplantation of stem cells from a CCR5 delta32 donor [9,10].

A primary focus of HIV research is to develop strategies that prolong protection and target latent HIV+ cells. Sustained ART-free remission is a well-accepted and effective strategy to control HIV infection. Recent studies have demonstrated that a long-acting (LA) antiretroviral (ARV) delivery system would boost the HIV treatment strategy [11,12,13,14]. The ARV nanoparticle-based delivery system has been shown to enhance drug solubility, stability, biodistribution, pharmacokinetics, efficiency, and concurrent drug safety due to reduced ARV-associated side effects [15,16,17]. Even though LA ARV nanoparticles (NPs) are promising as successful injectable LA antiretroviral, these delivery systems still need to show good tolerability. In the LATTE-2 trial, the cabotegravir plus rilpivirine LA injection group reported a higher incidence of grade 3–4 adverse events than the oral comparative treatment group [18]. The ECLAIR study of LA ARVs [19,20] revealed a significantly prolonged sub-therapeutic residual drug tail, placing patients at increased risk of contracting HIV and resistance development [21]. The emergence of ARV resistance would limit future treatment options in those patients treated with LA ARV NPs. 

Even though suppressing plasma viral load with ART has improved HIV treatment, it has been unsuccessful in immune restoration. Recovery from HIV needs the reestablishment of immunity against HIV. The research against HIV is facing challenges related to immune reconstitution failure. Various HIV vaccination strategies have failed so far because vaccine-induced inflammatory response and activation of CD4+ T cells (preferential HIV target) have enhanced the risk of infection propagation [22,23,24]. The inherent characteristic feature of HIV to target immune cells has been among the unique challenges faced in developing a prophylactic vaccine against HIV infection [22,25]; therefore, an effective HIV prophylactic strategy also needs to elicit protective anti-HIV immunity without further inducing inflammation and immune activation; however, the halted HVTN702 clinical trial emphasizes that unconventional and novel strategies are urgently needed to achieve an “HIV functional-cure” [26].

Studies have shown that HIV infection induces terminal differentiation of effector (E) T populations causing progressive E population reduction and immune exhaustion, promoting activation-induced cell death [27]. HIV infection strongly compromises the immune system, and immune impairment results in its inability to respond to other pathogens, progressing rapidly to AIDS. Ideally, a “protective immune reconstitution” promotion by stimulating both humoral and cellular immune responses could be a potential alternative to prevent and control HIV infection. The adaptive immune system plays a critical role in protecting against HIV infection. Studies have shown that promoting T-cell-based immunity, specifically cytotoxic T lymphocyte (CTL) stimulation [28,29], would promote effective neutralizing antibody production and synergistically protect against active and latent HIV infection. Thus, to promote a “protective anti-HIV immune” response, our strategy combines two HIV treatment strategies, i.e., ART and antibody-mediated HIV-entry blocking (Figure 1). We hypothesize that this combination strategy would effectively suppress HIV viremia and could promote protective anti-HIV immunity. The cumulative effect could potentially induce a “functional cure.” 

## 2. Materials and Methods

### 2.1. Materials

Acid terminated PLGA (75:25, Mn = 4000–15,000 Da), polyvinyl alcohol (PVA), dimethyl sulfoxide (DMSO), potassium dihydrogen phosphate (KH_2_PO_4_), dichloromethane (DCM), interlukin-2 (IL-2), phosphate-buffered saline (PBS), and methanol were all purchased from Sigma-Aldrich (St. Louis, MO, USA). Poly(lactide-co-glycolide)-block-poly(ethylene glycol)-succinimidyl ester (PLGA-PEG-NHS) and Methoxy Poly(ethylene glycol)-b-Poly(lactic-co-glycolic acid; PEG-PLGA (5000:15,000 Da, 75:25 LA:GA) were purchased from Akina, Inc. (West Lafayette, IN, USA). Pluronic F127 (the stabilizer) was obtained from D-BASF (Edinburgh, UK). 

The ARV drug, i.e., dolutegravir (DTG, 98% purity), was purchased (BioChemPartner Co., Ltd., Shanghai, China), and tenofovir alafenamide (TAF, 100% purity) was a generous gift from Gilead Sciences Inc. (Foster City, CA, USA) under the MTA agreement. Internal standards, i.e., tenofovir-d6 (TFV-d6), TAF-d5, tenofovir-diphosphate-d6 (TFV-dp-d6), and dolutegravir-d4 (DTG-d4) were purchased from Toronto Research Chemicals Inc. (Toronto, ON, Canada).

Roswell Park Memorial Institute (RPMI) 1640 with L-glutamine medium, Dulbecco’s Modified Eagle Medium (DMEM) high glucose medium (HiDMEM), and 100× antibiotic-antimycotic (AA) were purchased from ThermoFisher Scientific (Waltham, MA, USA), whereas fetal bovine serum (FBS) was from VWR International (Radnor, PA, USA). All the chemicals were used as received.

### 2.2. Primary Cells and Cell Lines

The TZM-bl cell line was obtained from the National Institutes of Health (NIH) acquired immunodeficiency syndrome (AIDS) reagent program. A JC53-bl (clone 13)/HeLa cell line phenotypically similar to HIV infecting cell type (stably overexpresses CD4 and CCR5 receptor) was used as an HIV-infection indicator cell type. TZM-bl cells were maintained in a complete DMEM medium (HiDMEM medium supplemented with 10% FBS and 1× AA) as standard protocol [14,15].

The human peripheral blood mononuclear cells (hPBMCs) were purchased from AllCells^®^ (Alameda, CA, USA) and maintained in complete RPMI (RPMI 1640 medium supplemented with 10% FBS, 1 × AA, and 50 U/mL IL-2 (Sigma-Aldrich, St. Louis, MO, USA)). The XF27/28/CCR5 43E2-AA hybridoma cell line (producing modified CCR5 mAb, referred to as ‘xfR5 mAb’; ATCC PTA-4054) was maintained in RPMI supplemented with 10% FBS and 1 × AA [32].

### 2.3. HIV Strain

HIV_ada_ and HIV-1_NL4-3_ viruses were obtained from the NIH AIDS research program and propagated further by the following standardized method [33]. The TCID_50_ was evaluated by standardized p24 ELISA assay using ZeptoMetrix^®^ HIV Type 1 p24 Antigen ELISA kit (Buffalo, NY, USA) on PBMCs received from healthy donors following the manufacturer’s protocol. HIV-1_NL4-3_ virus infectivity and TCID_50_ were evaluated on TZM-bl cells by Steady-Glo™ Luciferase Assay System (Promega; Madison, WI, USA) following standardized protocol [14,15,34,35,36].

### 2.4. Production, Purification, and Characterization of xfR5 mAb

The modified high-affinity xfR5 mAb was produced and isolated from a hybridoma cell line, i.e., XF27/28/CCR5 43E2-AA (PTA-4054; ATCC repository) [32], by following the published method with modifications as described below [37]. Briefly, XF-CCR5 28/27 43E2AA hybridoma cells were seeded (at 10^6^/mL concentration) and maintained in antibody production inducing media, i.e., RPMI medium with 1 × AA, for several days until 50% cells were found to be compromised (cell death). The supernatant with soluble xfR5 mAb was harvested by pelleting out dead cells and debris. The soluble xfR5 mAb from the supernatant was isolated using HiTrap™ Protein-A HP prepacked column (GE Healthcare; Chicago, IL, USA) following standard manufacturer’s protocol. The purity and concentration of the xfR5 mAbs were determined by the SDS-PAGE method and BCA assay using Pierce™ BCA Protein Assay Kit, following manufacturers’ protocol. The xfR5 binding was evaluated based on the standard curve (linear regression analysis) from a known concentration of IgG4 isotype control mAb, as xfR5 is recombinant human CCR5 mAb with IgG4 isotype backbone [32].

### 2.5. ARV-Loaded NPs Formulation

The targeted nanoformulation was fabricated by following multiple steps. First, NHS functionalized DTG+TAF (D+T) loaded nanoformulation was obtained following the modified oil-in-water emulsions phase inversion method [14,15]. Briefly, in the DCM organic phase, PLGA, PLGA-PEG-NHS, PEG-PLGA, and PF127, along with TAF and DTG, were dissolved at 1:1:2:2:4:4 ratios. TAF (at comparative ratio 2) was dissolved in PBS (aqueous phase). The aqueous phase was added dropwise to the organic phase under constant stirring conditions. The water-in-oil (w-o) emulsion obtained was sonicated as described below and then added dropwise to the three times higher volume of 1% PVA solution (aqueous phase) under high-speed stirring conditions. The above w-o-w emulsion was immediately probe-sonicated for 5 min on ice (setting: 90% Amplitude; pulse 0.9 cycle/bursts) with the help of a UP100H ultrasonic processor (Hielscher Inc. Mount Holly, NJ, USA). The organic phase from the o-w emulsion was completely evaporated overnight (O/N). The NHS functionalized D+T NPs were desiccated by lyophilization using Millrock LD85 lyophilizer (Kingston, NY, USA). The complete formulation method was carried out under the hood to maintain sterility during fabrication.

### 2.6. ARV NP Functionalization

The modified anti-CCR5 (xfR5) mAb was conjugated to the NHS functionalized D+T NPs by an amide bond [38]. Briefly, NHS ester-D+T NPs were dissociated in PBS (pH 7.4) as the NHS esters (4–5 h half-life of at pH 7.4). The xfR5 mAb in PBS has protonated amine groups. The protonated xfR5 mAb were added to NHS ester-D+T NPs at a 1:10 weight ratio under constant stirring at room temperature (RT), and the reaction proceeded for 2 h at RT. Immediately after the reaction, the unbound xfR5 mAb was washed off by dialysis using Float-A-Lyzer™ G2 Dialysis Devices (Thermo Fisher Scientific; Santa Clara, CA, USA) in cold PBS supplemented with 10mM hydroxylamine to quench any non-reacted NHS groups present on the xfR5-D+T NPs surface. Following three cold PBS buffer exchanges, xfR5-D+T NPs were collected and stored at 4 °C. The BCA assay evaluated the concentration of xfR5 mAb conjugated on the xfR5-D+T NPs surface. The PBS and D+T NPs were run in parallel during the BCA assay to obtain background values. The background values were subtracted from the mAb NPs absorbance value to obtain normalized mAb concentration of mAb on NPs.

### 2.7. CCR5 Targeted ARV NP Characterization

The physicochemical properties of the D+T NPs were evaluated based on dynamic light scattering (DLS), Fourier transform infrared (FT-IR) spectroscopy, and scanning electron microscopy. The size, surface charge, and polydispersity index (PDI) of the D+T NPs were determined by a ZetaPlus Zeta Potential Analyzer instrument (Brookhaven Instruments Corporation; Holtsville, NY, USA) following standardized methodology [14,15]. The DLS analysis was used to determine the D+T NPs size and polydispersity index (PDI), i.e., size homogeneity and distribution pattern. The zeta potential analysis identified the surface charge density on the D+T NPs. The D+T NP’s surface NHS functionalization and xfR5 mAb binding via amide bond was evaluated on FT-IR spectroscopic analysis following a previously published method [39]. Briefly, the spectra of each sample in powder form were collected in the range 600–4000 cm^−1^ using 25 scans at a resolution of 4 cm^−1^ with% transmittance intensity mode and Happ–Genzel function apodization under IRPrestige-21 Fourier transform infrared spectrometer instrument (Shimadzu; Columbia, MD, USA). The data were analyzed using LabSolutions IR software (Shimadzu; Columbia MD, USA). By scanning electron microscopy, the morphology and shape of the D+T NPs were evaluated [40]. Briefly, D+T NPs were deposited on Whatman^®^ Nuclepore Track-Etch Membrane (~50 nm pore size) and air-dried for one day at RT under a chemical hood. The air-dried NPs membrane was sputter-coated with a thin layer (~3–5 nm thick) of chromium and imaged under a Hitachi S-4700 field-emission SEM (New York, NY, USA).

The % drug entrapment efficiency (%*EE*) of DTG and TAF in D+T NPs and xfR5-D+T NPs were evaluated by high-performance liquid chromatography (HPLC) instrument by following the published methodology [34,41,42]. Briefly, 1 mg of D+T NPs dissociated in 50 µL DMSO, and an appropriate amount of mobile phase (25 mM KH_2_PO_4_ 45%:ACN 55%) was added to obtain a 10% DMSO concentration in the final injection volume (20 µL). The same procedure was followed for the standard curve evaluation to prepare the D+T standard solutions (with each drug concentration from 0.5 to 0.0019 mg/mL). The chromatography separation was performed under an HPLC instrument (Shimadzu Scientific Instruments; Columbia, MD, USA) equipped with SIL-20AC auto-sampler, LC-20AB pumps, and SPD-20A UV/Visible detector, using Phenomenex^®^ C-18 (150 × 4.6 mm, particle size 5 μm) column (Torrance, CA, USA), under isocratic elution process. The mobile phase was maintained at 0.5 mL/min flow rate, temperature: 25 °C, and detection at 260 nm (retention time of 4 min for TAF and 6.3 min for DTG). The quantification of the drug was determined by evaluating the peak area under the curve (AUC) analysis at their respective retention time. The amount of TAF and DTG loaded in the D+T NPs was analyzed based on the standard curve construction (linear correlation, r^2^ ≥ 0.99) respective from TAF and DTG standard concentration ranges from 0.5 mg/mL to 0.0019 mg/mL. The HPLC instrument illustrated inter-day and intra-day variability of <10%. The % encapsulation efficiency (%*EE*) of each drug in the D+T NPs batch was estimated by Equation (1). The data are presented as mean ± standard error of the mean (SEM) of three D+T NPs batches (*n* = 3).
(1)$$\%EE=\frac{Amount\;of\;drug\;entraped\;in\;NPs\;\left(\mathrm{mg}\right)}{Amount\;of\;drug\;used\;for\;encapsulation\;\left(\mathrm{mg}\right)}\times 100$$

### 2.8. Antibody Binding and Binding Affinity Evaluation

The binding affinity of isolated xfR5 mAb and xfR5-D+T NPs compared to wild-type CCR5 mAb (rabbit anti-human mAb; Bioss Inc.; Woburn, MA, USA) was evaluated by flow cytometry. The xfR5 mAb and wild-type Ab were conjugated with Cy3 dye by Cy3 NHS Ester Mono-Reactive CyDye (GE Healthcare; Chicago, IL, USA), following the manufacturer’s protocol. The Cy3 dye to xfR5 mAb binding ratio was evaluated based on regression analysis of respective standards (i.e., 0.5 to 0.00625 mg/mL) data obtained from UV/vis spectroscopy and BCA assay. The batches with a 3:1 ratio of Cy3 dye to xfR5 mAb (Cy3-xfR5 mAb) and wild-type CCR5 mAb were considered for further studies. To study binding affinity by flow cytometry, Cy3-xfR5 mAb conjugated D+T NPs (Cy3-xfR5-D+T NPs) formulations were fabricated and characterized as described for xfR5-D+T NPs. Three independent batches were obtained and evaluated using the standardized formulation for further studies.

For the binding affinity of Cy3-xfR5 mAb and Cy3-xfR5-D+T NPs evaluation, TZM-bl cells (10^5^/well) and phytohemagglutinin (PHA, at 5 µg/mL) activated PBMCs (10^6^ cells/well) were treated with Cy3-xfR5 mAb and Cy3-xfR5-D+T NPs at different concentrations (20, 10, 1, 0.1, 0.1 µg/mL of xfR5 concentration) O/N at 37 °C and 5% CO_2_ atmosphere. In parallel, the wild-type Cy3-CCR5 mAb treated PBMCs at the same concentration and considered a control group. PHA is known to induce latent HIV-infected primary T-cells’ reactivation [42]; therefore, to achieve latent HIV-infected primary T-cells phenotype (promote CD2+ T-cells), PHA-activated PBMCs were used. The treatment was washed off thrice with 1% BSA in PBS (PBA) solution by centrifugation (220× *g* for 10 min at 4 °C). As HIV-1 primarily infects CD4 T-cells, the binding affinity of xfR5 mAb and xfR5-D+T NPs compared to wild type anti-CCR5 mAbs to CD4+ T cells was evaluated by labeling the above-treated cells with anti-CD4 AlexaFluor700 mAb (Table 1) for 20mins at RT (at 1:100 dilution) and washed thrice with PBA. Similarly, to evaluate binding affinity with the latent population (CD2+ T-cells) and monocytes (CD68+ T-cells), treated PBMCs (as described above) were labeled with anti-CD68 APC mAb and anti-CD2 Pacific Blue, respectively (Table 1). In parallel, unstrained and single channel marked labeled PBMCs were considered for flow cytometry setting and gating strategy. The above marker antibody labeled treated and untreated cells were fixed for 20 min with 4% PFA at 4 °C and washed twice with PBA. The binding of Cy3-xfR5 mAb and Cy3-xfR5-D+T NPs to respective T-cell types was detected and evaluated by the BD LSRII flow cytometer instrument (BD Biosciences; San Jose, CA, USA) and Flowjo software v10 (BD, Franklin Lakes, NJ, USA), respectively. Appendix A details the complete gating strategies. Each experiment was performed on three healthy independent donors PBMCs (*n* = 3). The binding affinity was calculated based on Michaelis–Menten’s non-linear fitting analysis of obtained mean ± SEM (standard errors of means) data.

### 2.9. Immunophenotype Study

Immunophenotype variation upon xfR5-D+T NPs treatment compared to xfR5 mAb in uninfected (mimicking pre-exposure prophylaxis, PrEP condition) and HIV-1_ADA_-challenged/infected PHA-activated PBMCs was evaluated by flow cytometry. Briefly, for immunophenotype evaluation in the absence of infection (during PrEP application), PBMCs (10^5^ cells/well) were treated (day 0), respectively, with xfR5 mAb and xfR5-D+T NPs (at 20 µg/mL of xfR5 concentration) for 96 h at 37 °C and 5% CO_2_ atmosphere for 4 days (detailed in Appendix A). As a control group and to compare activated PBMCs immunophenotype, PHA-activated PBMCs (10^5^ cells/well) were maintained for 4 days [36]. For the immunophenotypic study during the HIV challenge (Appendix A), the respective cells were treated on days 0 and 1. The treated cells were challenged with HIV-1_ADA_ virus (MOI: 0.1) for 16 h, followed by infection wash-off (three-time fresh media wash), and the cells were maintained in a fresh medium for 4 days. For the HIV-infection cell treatment condition (Appendix A), the cells were infected on day 0 with the HIV-1_ADA_ virus, as mentioned above, after washing off the PHA stimulation (day^−1^). After washing off HIV infection, cells were treated, respectively, with xfR5 mAb and xfR5-D+T NPs (20 µg/mL of xfR5 concentration) for 4 days. All cells were PAH-stimulated on day^−1^. On the respective days (days 0, 1, and 4), cells were washed thrice with cold PBA solution by centrifugation (400× *g* at 4 °C) and incubated with marker mAbs against T-lymphocytes (CD3), helper T-cells (CD4), cytotoxic T-cells (CD8), memory T-cells (CD45RO), transition T-cells (CCR7), activated T-cells (CD69), intermediate memory T-cells (CD27), and HIV latently infected T-cells (CD2) markers (as detailed in Table 1), for 20 min at RT (at 1:100 dilution). The cells were again washed with PBA, fixed for 20 min with 4% PFA at 4 °C, and rewashed thrice with PBA. The immunophenotype was evaluated by flow cytometry. The untreated and unstimulated were considered as the initial cellular immunophenotypic profile. Three independent studies have been performed on three healthy donors’ PBMCs. The data are presented as mean ± SEM was obtained from three independent donors.

### 2.10. Intracellular Kinetics Experiments

The intracellular uptake and retention kinetics of D+T NPs and D+T solution were evaluated by liquid chromatography with tandem mass spectrometry (LC-MS/MS) analysis following a standardized method [14,42,43]. Briefly, TZM-bl cells (10^4^ cells/well) were seeded in the 24-well plate with the complete HiDMEM medium. Following O/N cell adherence, the respective cell groups were treated with D+T NPs and D+T solution at 10 µg/mL concentration of each drug, i.e., DTG and TAF. For uptake experiments, at respective time points (i.e., 1, 6, 18, and 24 h), the treated cells were washed twice with warm PBS and detached by Trypsin-EDTA (25%; Thermo Scientific, Oklahoma City, OK, USA), washed twice with PBS. One set of untreated detached cells was counted at each time point to determine cell count at the respective time point. The cells were air-dried under a biosafety cabinet. The air-dried samples were then lysed with 70% methanol and stored at −80 °C until analysis. Drug-retention experiments, the adhered TZM-bl cells were treated with xfR5-D+T NP and D+T solution, respectively, for 24 h and washed thrice with warm 1 × PBS. The washed treated cells were in fresh complete HiDMEM medium until respective time-points (i.e., 1, 6, 24, and 72 h after wash, which corresponds to 25, 30, 48, and 96 h, respectively) after treatment time points). The cells were rewashed with PBS at this time point, detached, lysed, and stored following the same method as explained above. The samples were analyzed using the LC-MS/MS method described in the section below.

For the intracellular DTG, TAF, TFV, and TFV-dp drug-kinetics evaluation by LC-MS/MS instrument, the cell lysates were centrifuged (21,952× *g* for 5 min at 4 °C), and the supernatant was collected. To an aliquot of 100 µL supernatant, 300 µL of internal standard spiking solution (10 ng/mL each of DTG-d4, TAF-d5, TFV-d6, and 100 nmol/mL of TFV-dp-d6 in ACN) was added and vortexed. The samples were then dried at 45 °C under the stream of nitrogen and reconstituted with 100 µL of 50% acetonitrile. The drug and metabolites were quantified using LC-MS/MS instrument from the same sample.

For TAF, TFV, and DTG estimation, the similar conditions that our group previously published were used with minor modification [43]. One µL of the processed sample was injected into LC-MS/MS and operated in positive mode. The chromatographic separation was carried out using the Restek Pinnacle DB Biph column (2.1 mm × 50 mm, 5 µm) with 0.5% formic acid in water and 0.1% formic acid in ACN (48:52 *v*/*v*) mobile phase. The calibration range for all the analytes was 0.01 to 50 ng. 

For the quantification of TFV-dp, Phenomenex Kinetex C18 (75 × 4.6 mm, 2.6 µm) column was used with an isocratic mobile phase (10 mM ammonium acetate pH 10.5: ACN (70:30) at a flow rate of 0.25 mL/min. The dynamic calibration range was from 0.01 to 100 ng. The LC-MS/MS system consists of an Exion HPLC system (Applied Biosystems, Waltham, CA, USA) coupled with AB Sciex 5500 Q Trap with an electrospray ionization (ESI) source (Applied Biosystems, Waltham, CA, USA) was used in positive ionization mode. The retention time of TFV-dp was 2.1 min, and the runtime for each sample was 3.5 min. The average inter-day and intra-day variability were <15%, corresponding to the FDA bioanalytical guidelines [44].

### 2.11. In Vitro Cytotoxicity Experiments

The comparative in vitro cytotoxicity of D+T NP vs. D+T solution was evaluated using the TZM-bl cell line and CellTiter-Glo^®^ luminescent assay method, as described previously [45]. Briefly, the TZM-bl cells (10^4^ cells/well) in complete DMEM medium and PBMCs (10^5^ cells/well) in complete RPMI medium, were treated in triplicate, respectively, with D+T NP or D+T solution, at different concentrations (20, 10, 1, 0.1, 0.01 µg/mL each drug concentration) for 96 h. Similarly, the 5% DMSO-treated cells and 1×PBS (treatment equal volume) treated cells were the positive and negative control, respectively. The cytotoxicity was evaluated by the CellTiter-Glo^®^ luminescent-cell viability assay kit (Promega; Madison, WI, USA) following manufacturer protocol. The luminescence intensity was read from the Synergy II multi-mode reader with Gen5TM software (BioTek; Winooski, VT, USA). The untreated control cells were considered 100% viable. The percentage cytotoxicity (*%Cytotoxicity*) was calculated as % normalized viability against the untreated (negative) control group. The experiment was carried out on three independent batches of D+T NPs and D+T solution. The result represents the mean ± SEM of three independent batches of studies. The *%Cytotoxicity* was evaluated by following Equation (2):(2)$$\%Cytotoxicity=\frac{RLU_{Untreated}-RLU_{Treated}}{RLU_{Untreated}}\times 100$$

### 2.12. In Vitro Protection Experiments 

The comparative in vitro prophylaxis (PrEP), i.e., protection study between D+T NP vs. D+T solution against HIV-1_NL4-3_, was performed on TZM-bl cells and peripheral blood mononuclear cells (PBMCs) was evaluated by following standardized method [14,45]. Briefly, TZM-bl cells (10^4^ cells/well) and PBMCs (10^5^ cells/well) were seeded in 96-well plate and were treated with different concentrations of D+T (20, 10, 1, 0.1, 0.01 µg/mL each drug concentration) either as D+T NP or as D+T solution. Untreated/uninfected cells and untreated/infected cells were considered negative and positive controls. After 24 h of treatment, the TZM-bl cells were infected with HIV-1_NL4-3_ virus (multiplicity of infection, MOI: 1) for 8 h, whereas PBMCs received HIV-1_ADA_ virus (MOI: 0.1) for 16 h. At the respective time points, the inoculated and control cells were washed with warm PBS (thrice). The TZM-bl cells and PBMCs were then maintained in fresh complete HiDMEM medium and complete RPMI, respectively, for 96 h. In the case of TZM-bl cells, the HIV-1 infectivity was evaluated by the luminescence intensity following Steady-Glo^®^ luciferase assay (Promega; WI, USA), following company specified methodology. Based on relative luminescence units (RLU), the luminescence intensity was read on the Synergy HT Multi-Mode Microplate Reader (BioTeck; Winooski, VT, USA). The % HIV-1 infection was calculated by following Equation (3):(3)$$\%\;\mathrm{HIV}\;\mathrm{protection}=\frac{\left(RLU_{untreated/infected}-RLU_{treated/infected}\right)}{RLU_{untreated/infected}}\times 100$$

However, in the case of PBMCs, the HIV infectivity was evaluated based on optical density (*OD*) produced by p24 antigen using ZeptoMetrix^®^ HIV Type 1 p24 Antigen ELISA kit and following the standardized manufacturer’s protocol. The *OD* values as HIV infectivity was read on the Synergy HT Multi-Mode Microplate Reader. The % HIV-1 infection was calculated by Equation (4):(4)$$\%\;\mathrm{HIV}\;\mathrm{protection}=\frac{\left(OD_{untreated/infected}-OD_{treated/infected}\right)}{OD_{untreated/infected}}\times 100$$

For TZM-bl cells, data were collected from three independent experiments performed with three different batches of D+T NP and D+T solution (each performed in duplicate). For PBMCs, the data were obtained after treating three different healthy donors’ PBMCs (each performed in triplicate) at independent time. Finally, the selectivity index (SI) was evaluated by following Equation (5):(5)$$SI=\frac{CC_{50}}{IC_{50}}$$
where ‘*CC*_50_’ (cytotoxic concentration at 50%) and ‘*IC*_50_’ (50% inhibitory concentration) were evaluated from the above-described in vitro cytotoxicity and protection study.

### 2.13. Statistical Analysis

All study results presented are expressed as mean ± SEM of the obtained data from at least three independent experiments. The *CC*_50_ value was determined by non-regression curve fitting based on log (DTG or TAF concentration) vs. normalized luminescent (three-parameter logistic fits) of cytotoxicity response curves. In contrast, the *IC*_50_ value was evaluated by fitting the non-regression inhibitory curve of log [DTG] vs. normalized TZM-bl luminescence (three-parameter logistic fits) luminance values. The analysis of variance (ANOVA) method was used to determine significant differences between treated (D+T NP and D+T solution) vs. control groups at a *p*-value ≤ 0.05. All the statistical analysis presented was determined by GraphPad Prism 7 software (La Jolla, CA, USA).

## 3. Results

### 3.1. CCR5 Targeted cARV-Loaded NPs Characterization

By using standardized water-in-oil-in-water (w-o-w) interfacial polymer deposition method [14,15], N–hydroxysuccinimide (NHS) functionalized DTG and TAF loaded NPs (D+T NP) were formulated (Figure 1). The DTG and TAF drugs were loaded at a 1:1 ratio in the modified CCR5 mAb (xfR5-D+T NP). Well-defined DTG+TAF NP (NHS-D+T NP) was formed as a result. Dynamic light scattering (DLS) size distribution analysis demonstrated D+T NPs size averaged 198.7 ± 10 nm with uniform size-distribution (poly-dispersity index, PDI: 0.153 ± 0.008) (Table 2). The xfR5 mAb were covalently conjugated on NHS-D+T NP (xfR5-D+T NP) by replacing the NHS group. The xfR5 mAb binding on xfR5-D+T NP caused a slight increase in size (212.6 ± 20.7 nm). The shape and surface properties evaluation by scanning electron microscopy reflected that NHS-D+T NPs obtained were uniform and smooth-surfaced spherical particles (Figure 2A), and xfR5 mAbs surface conjugation did not cause changes in the particle’s morphology (Figure 2B). The xfR5 mAb binding on the D+T NP surface caused NP’s surface charge suppression from −28.15 ± 1.9 mV to −17.77 ± 1.9. This surface charge reduction could be attributed to several factors, such as charge masking by the ions from the PBS buffer or xfR5 mAb surface decoration, or both. The polymeric NP encapsulation enhanced drug encapsulation efficiency (%*EE*) of 58.5 ± 5.2% and 55.7 ± 3.9% for DTG and TAF, respectively (Table 2).

The xfR5 mAb binding on xfR5-D+T NPs quantified by bicinchoninic acid (BCA) protein assay estimated ~3.7 ± 0.52 µg of xfR5 mAb per mg xfR5-D+T NPs (Table 2). The xfR5 mAb binding was further confirmed by FT-IR analysis, as shown in Figure 2C. The NHS ester bands at 1695–1818 cm^−1^ (aromatic C=O stretching) and 967 cm^−1^ (–CH wagging band) of PEG-PLGA-NHS polymer on NHS-Blank NPs and NHS-D+T NPs, confirmed free NHS functional group on the NP surface. The disappearance of these –NHS bands in xfR5-D+T NP proved the replacement of the NHS ester group. Additionally, the shift of primary aliphatic amine NH stretching (3350 cm^−1^ and 1420 cm^−1^) and –CH stretching (2947 cm^−1^), along with primary amide I and II band presence (at 1640 and 1540 cm^−1^, respectively), confirmed the covalent amide (–CONH–) bond formation. Other significant bands, C–O–C stretching (1000–1300 cm^−1^) and broad O–H stretching band (3000–3500 cm^−1^) confer xfR5-D+T NPs surface, which has open PLGA & PEG surfaces along with covalently bound xfR5 mAb. Based on these results, xfR5 mAb is not densely packed on xfR5-D+T NP and has enough free space to avoid steric hindrance during targeted CCR5 receptor binding. 

Overall, the enhanced electron density and BCA protein quantification validated the binding of xfR5 mAbs on the xfR5-D+T NPs. In contrast, the amide bond (NHCO) conferred a covalent bond between xfR5 mAb and PEG on the NP surface. Aside from the amide bond, the O–H stretching at 3000–3500 cm^−1^ confirms that the free surface on NP has predominant PEG-PLGA composition; therefore, the xfR5 mAbs are spread on the PEG-covered NP surface with a relatively good space between molecules to avoid steric hindrance.

### 3.2. Binding Affinity

The antibody binding affinity was estimated using flow cytometric analysis, and the data are expressed as equilibrium dissociation constant, K_d_. The binding affinity (K_d_) of xfR5-D+T NP compared to naïve xfR5 mAb and wild-type CCR5 mAb on TZM-bl cells (Table 3) demonstrates that xfR5 mAb displayed approximately 10-fold higher affinity (0.251 ± 0.15 nM) compared to wild-type CCR5 mAb (2.023 ± 0.655 nM). The simultaneous-multiple xfR5 mAbs of xfR5-D+T NP (5.7 ± 2.7 nM) interaction with CCR5 receptors of cells improved the sensitivity by approximately 6.6 times (Appendix A and Table 3). Similarly, xfR5-D+T NPs showed enhanced binding with the primary CD4+ T-cells (4.31 ± 1.47 nM), as reflected from the Kd value, i.e., approximately Kd value was ~5 times lower than the xfR5 mAb (20.87 ± 10.65 nM); therefore, multipoint interaction of single xfR5-D+T NP via multimeric xfR5 mAb on NP significantly improved binding affinity compared to xfR5 mAb di-valent interaction (Appendix A).

The binding affinity of xfR5-D+T NPs with other CCR5+ immune cell types such as cytotoxic T-cells (CTLs, CD8+), dendritic T-cells (CD2+), and monocytes (CD68+), were evaluated using a similar method (gating strategy, Appendix A). The comparative binding affinity demonstrated that xfR5-D+T NPs had a significantly enhanced binding affinity with memory CD8+ T-cells, approximately 25 times higher affinity (*p* < 0.05) compared to naïve xfR5 mAb (Table 2). Additionally, xfR5-D+T NP binding with CD2+ T-cells and CD68+ T-cells displayed slightly enhanced but non-significant differences in binding affinity compared to naïve xfR5 mAb.

### 3.3. Intracellular Drug Kinetics

The sustained release property of xfR5-D+T NP compared to D+T solution was evaluated following the intracellular uptake/release and retention kinetics of DTG, TFV, and TFA-dp (active-metabolite) by non-compartmental analysis using Phoenix WinNonlin 8.1 software. The result demonstrated (Table 4) that the maximum concentration (C_max_) and the area under the concentration-time curve (AUC_all_) of TAF after xfR5-D+T NP treatment were 12.9 and 72.4 times higher than TAF solution treatment. DTG demonstrated comparatively higher C_max_ and AUC_all_, (110.9 and 254.5 times higher, respectively). These parameters showed a non-significant difference for intracellular TFV and TFV-dp between xfR5-D+T NP and D+T solution. From the C_max_ and AUC_all_ data of TAF and DTG, it is evident that the nanoformulation enhances cellular uptake compared to the naïve drugs in solution. In terms of retention, NP formulation demonstrated prolongation in elimination half-life (t_1/2_) of 11.6 and 4.4 times for TAF and DTG compared to naïve drugs in solution.

### 3.4. Cytotoxicity and HIV Protection Study

A comparative cytotoxicity study was performed using TZM-bl cells and primary human peripheral blood mononuclear cells (hPBMCs) (Table 5). The % viability was calculated based on untreated controls (normalized using Equation (2)). In primary hPBMCs, xfR5-D+T NP treatment demonstrated a non-significant change in *CC*_50_ compared to xfR5 mAb (Appendix A). Overall, in vitro cytotoxicity results in TZM-bl cells and hPBMCs suggest that nano-encapsulation does not pose added toxicity and improves cell viability. The experiments revealed that all the tested nanoformulations (xfR5-D+T NP, xfR5 NP, and D+T NP) were as safe as xfR5 mAb for cellular application. These results indicated that PLGA-based CCR5 targeted ARV drugs loaded NP does not induce significant cytotoxicity.

The dual protection by xfR5-D+T NP due to its multimeric xfR5 mAb blocking and prolonged ARV release was anticipated to improve the half-maximal inhibitory concentration (*IC*_50_) compared to xfR5 mAb and D+T NPs alone. The protection study in TZM-bl cells demonstrated that xfR5-D+T NP improved *IC*_50_ by 528 and 5.5 times compared to D+T NPs alone and xfR5 mAb alone, respectively (Table 5). Similarly, in primary hPBMCs, xfR5-D+T NPs lowered the *IC*_50_ by 12, 5.5, and 3.4 times compared to naïve xfR5 mAb, xfR5 NP alone D+T NP alone, respectively (Table 5 and statistical test in Appendix A). These results support the fact that multimeric mAb-coated nanoformulation induces multivalent interaction, which may have enhanced HIV protection compared to xfR5 mAb alone (Appendix A). Finally, using the selectivity index to evaluate the cytotoxicity versus efficacy demonstrated the potential of xfR5-D+T NPs. These results demonstrate that xfR5-D+T NP significantly (*p* < 0.05) improved the therapeutic index of both individual therapeutic approaches alone.

### 3.5. Immunophenotype during In Vitro Short-Term Prophylaxis and HIV Treatment Study

The immunological potential of targeted ARV-loaded nanoformulation was evaluated on the PBMCs isolated from healthy donors. The immune-differentiation pattern of T-cells during prophylaxis application and HIV treatment were evaluated using flow cytometry after treating uninfected hPBMCs with xfR5-D+T NP or xfR5 mAb, along with respective controls. The gating strategy for the immunophenotypic study is described in Appendix A. Based on the expression level of CCR7, CD27, and CD45RO receptors on T-cells, five distinct T-cell subpopulations were determined, i.e., naïve cells (CCR7+ CD27+ CD45RO−), central memory cells (CM, CCR7+ CD27+ CD45RO+), transition memory cells (TM, CCR7− CD27+ CD45RO+), effector memory cells (EM, CCR7− CD27− CD45RO+), and effector cells (CCR7− CD27− CD45RO−). The immunophenotype of the T-cell populations was determined over different time points and conditions. The time point considered were (i) day before phytohemagglutinin (PHA) activation (unstimulated T-cell population); (ii) PHA-activation (one day after PHA-activation); (iii) HIV infection (one day after HIV infection); as well as (iv) one and four days after xfR5-D+T NP or xfR5 mAb treatment (Figure 3).

The unstimulated hPBMCs was composed of 18%, 30.3%, 1.4%, 60.7%, and 36.8% of naïve, CM, TM, EM, and E T-cell subpopulations, respectively (Figure 3A). As expected, PHA stimulation significantly (*p* < 0.05) changed the immunophenotype of T-cell sub-populations stimulating the naïve and E sub-population (45.7% and 66.1%, respectively). EM population was significantly (*p* < 0.05) decreased (27.4%) during PHA stimulation. Emergence of activated cytotoxic T-lymphocytes (aCTLs, CD8+ CD69+) from minimal (0.66%) to 41.8% could also be attributed to PHA-activation (Figure 4). Further, the binding of xfR5-D+T NPs with different immune cell types than xfR5 NPs and xfR5 mAb (Appendix A) persists not only with CD4+ cells but binds with all cell types expressing CCR5 receptor.

The short-term ex vivo HIV protection study (4 days) demonstrated that xfR5-D+T NP significantly protected cells against HIV infection (Table 5), which could be attributed to a protective immunophenotype (Figure 3B). The xfR5-D+T NP resulted in increased CM, TM, and E sub-populations; however, compared to untreated, PHA-activated hPBMC, the xfR5-D+T NP significantly boosted TM and E sub-populations (Figure 3B). The aCTLs and T_h_ population demonstrated an inverse effect (Figure 4B). The aCTL sub-population after the initial spike (day 1) followed a decline. In contrast, the T_h_ population gradually increased over time (Figure 4A,B). 

The ex vivo HIV infection treatment and short-term (4-day) protection experiments were performed to predict the immunophenotypic differentiation pattern that the nanoformulated treatment could produce (Figure 3C). Results demonstrated that HIV infection mainly influenced naïve and CM sub-population (Figure 3C). In the presence of HIV infection, the naïve sub-population, after an initial increase (day 1, post-infection, PI), reverts to basal levels (day 4, after HIV infection), and the CM population showed a gradually increased population (day 4 PI). The xfR5-D+T NP treated population also showed enhanced naïve and CM sub-population during the initial active HIV infection stage (day 1 PI). It would be interesting to investigate the immunophenotypic differentiation pattern in treatment-experienced patients for long-term effects.

## 4. Discussion

Our primary aim was to investigate a potential treatment modality to investigate a strategy that could achieve ‘functionally cure’ HIV+ patients. These experiments demonstrated a strategy combining two different HIV-treatment approaches in a single nanoformulation with the sustained-release property. Combining anti-CCR5 HIV-entry blocking [3] and ARV-based HIV treatment [48]. We formulated xfR5-D+T NPs, a CCR5 receptor targeting ARV-loaded nanoformulation with the potential to block HIV entrance and promote cellular immunity against HIV infection, specifically in memory CD4+ T-cells as they commit themselves to provide antiviral immunity as latent HIV reservoir [34]. Alongside memory CD4+ T-cells, the myeloid lineage, such as monocytes/macrophages, are believed to be other potential HIV sanctuaries [35,36]. We hypothesize that the CCR5 mAb-targeted ARV nanoformulation combines two different strategies to ensure protection against HIV. First, CCR5 mAb docking on the CCR5+ cells blocks HIV entry and prevents new HIV infection, following a similar action mechanism as ARV maraviroc. Second, sustained intracellular ARV released from endocytosed NP degradation would provide intracellular protection against HIV to the naïve cell and latent HIV-infected cell population (Figure 1C). In addition, prolonged intracellular ARV levels and CCR5 blocking may promote anti-HIV immunological response. Thus, prolonged HIV protection and immunity-boosting could contribute to a possible “functional cure” strategy [37].

The only effective CCR5 receptor blocker that blocks HIV entry in lymphocytes and confirmed HIV cure is the CCR5 delta32 natural mutant genotype [9,10,49]. Other strategies, including maraviroc, ZFN, ribozymes, TALENs, short hairpin RNAs, and CRISPR/Cas9 nuclease, have been shown to block HIV entry but are not yet successful in conferring HIV cure [49]. A study by Platt et al. reported that approximately 7 × 10^2^ and 2 × 10^3^ CCR5 receptors per cell could lead to HIV infection [50]. To block HIV infection, at least 10^3^ blocker molecules are needed to occupy 7 × 10^2^ unbound CRR5 receptors/cell. The most effective CCR5 blocker molecule, maraviroc in vivo, does not accumulate at a concentration around the cell to block all CCR5 receptors. Additionally, ligand/CCR5 endocytosis increases CCR5 receptor turn-over dynamics, causing increased CCR5 density as viral load increases [51]. Incomplete blockage due to an insufficient number of molecules near CD4+ cells or increased CCR5 receptor density because of CCR5 receptor regulatory dynamics [52] is a major reason behind the failure of CCR5 blocking strategies other than the CCR5 delta32 natural mutant genotype approach.

To overcome these constraints, we strategically surface decorated xfR5-D+T NP with multiple xfR5 mAb to maximize CCR5 receptor blocking and prove a hindrance to viral binding to CCR5 receptors on T-cells. The xfR5 mAb immunospecifically binds to one or more types of CCR5-receptor [32,53]. It also effectively downregulates CCR5 surface expression by promoting its internalization. The patent report claims xfR5 mAb can inhibit or abolish HIV binding, enter/fusion, and replicating ability in CCR5 expressing cells [53].

Furthermore, compared to univalent interactions such as maraviroc, each xfR5-D+T NPs could effectively block multiple CCR5 receptors due to multimeric interaction of xfR5-D+T NPs via multiple xfR5 to CCR5 receptor interaction per cell. The enhanced binding affinity of xfR5-D+T NP could be attributed to the multivalent interaction strengthening the binding affinity between the target biomolecules on the NP and the receptors on the cell surface [54]. The theoretical calculation revealed that each mAb conjugated NP using the flexible PEG spacers could occupy ten mAb/receptor complexes in T-cells [54]; however, a single naïve mAb could only block one CCR5 molecule per T-cell [55]. Similar to the theoretical assumption, the xfR5-D+T NP practically resulted in a 7-times higher binding affinity than xfR5 mAb. The CCR5 co-receptor characteristically is expressed on CD4+ T-cells in peripheral blood [56]. In addition to CD4+ T cells, the CCR5 chemokine receptor is also expressed on CTLs (CD8+) [57], dendritic cells (CD2+) [58], and monocytes/macrophages (CD68+) [59,60], which also plays critical roles in HIV infection, propagation, and latency. The binding affinity of xfR5-D+T NP on other CCR5+ expressing cell types is essential to estimate the success of this targeted nanoformulated strategy. Our studies demonstrate the multimeric binding potency of xfR5-D+T NP on primary HIV susceptible cells type, especially memory Th (CD4+) and CTLs (CD8+) T-cell populations displayed ~5 and ~25 times higher binding affinity compared to naïve xfR5 mAb. The enhanced CCR5 expression during infection or PHA-activation promotes migration of activated effector and memory CTLs T-cells [57]. 

Additionally, the assembly of NPs size (~212 nm, with a weak surface negative charge) around target cells and strong cell surface interaction together pose a substantial steric hindrance to out-compete virion particle (sized ~145 nm [56]), thus minimizing their cellular binding (Figure 1A). Hence, steric hindrance strategically also contributes toward protection against viral entry. The above strategic twofold competitive hindrance by xfR5-D+T NPs may overcome the incomplete blockage constrain of other reported CCR5 blocking strategies. 

A study has shown that viral interaction with the CCR5 receptor promotes increased surface CCR5 density on the T-cell surface [57]. CCR5 receptor endocytosis and recycling rate increase compared to degradation [58]. Others have reported that blockage in HIV/CCR5 entry delayed HIV disease progression (observed in HIV+ long-term non-progressors populations) [59]. The NP interaction with the target T-cell generates strong forces following the same direction due to specific ligand-receptor binding and non-specific bonds (i.e., van der Waals, electrostatic and steric interactions). The increased membrane stiffness due to NP-multivalent receptor interaction thus possibly nullifies or overshades the receptor-mediated endocytic force as membrane tension increases [60]. Hence, NP-mediated interference with the ligand–receptor endocytosis process prolongs targeted NP retention [60]. This process could delay CCR5 receptor turn-over rate, suppressing CCR5-receptor upregulation, similar to long-term non-progressors [61]. Target-specific nanoformulation has several unique properties that block (physically and mechanically) HIV entry and delay or prolongs protection against new HIV infection.

Our strategy is to introduce internal protection against new infection or during reactivation of latently infected cells. Latent HIV+ cells, when activated, will produce new virion components (reverse transcriptase, integrase, and protease) to be packed into fresh virion particles to start a new HIV infection. Thus, the CCR5 targeted NPs were loaded with prescribed NRTI and integrase inhibitors to ensure intracellular protection [62,63,64]. The xfR5-D+T NPs upon endosomal uptake cause sustained release of ARVs in cell cytoplasm due to polymeric degradation; therefore, fresh HIV infection introducing new reverse transcriptase or integrase will immediately be inhibited by the intracellular NRTI and INSTI concentration (Figure 1, schematic diagram). The HIV protection assay results demonstrated the same possibility. The xfR5-D+T NP reduced IC50 value by 12 (hPBMCs) times compared to naïve xfR5 mAb. Further, our previous and present study confirms that nanoformulated ARV drugs promote a broad therapeutic index compared to naïve drugs [14,15,40,41,42]. 

HIV infection and ART induces different immunophenotypic profile. Studies have shown that HIV infection induces naïve T-cells proliferation, whereas ART causes depletion [65], which aligns with our observed (Figure 3C). Studies have shown that HIV-infected hPBMCs increased the naïve sub-population [66,67]. In contrast, xfR5-D+T NP induced HIV suppression and reverted the naïve sub-population to its basal level. We observed a reduced naïve T-cell population and an increased CM sub-population, indicating that targeted ARV nanoparticles promote effective memory against HIV by shifting the differentiation-induced population from the naïve ⇒ CM sub-population [68].

The memory T-cell population plays a vital role in promoting anti-HIV immunity and AIDS prognosis since these cells govern the functions of CTLs and B-cells, which regulates cellular and humoral immunity against HIV [69]. Persistent HIV viremia is known to drive the differentiation CM ⇒ EM sub-population [70,71], and the high EM sub-population may be responsible for the long-lasting and exhausted T-cell population in HIV+ patients [70]. The ART treatment partially restores the high CM sub-population and reduces the EM sub-population overtime to regain homeostasis disrupted during HIV infection [70]. Our results demonstrate that as treatment progresses, the xfR5-D+T NP treated population could significantly increase the CM sub-population and boost TM and EM sub-population (Appendix A); however, during active HIV infection, targeted treatment using xfR5-D+T NP or naïve xfR5 mAb maintained high E and aCTL sub-population to counter HIV infection. The E and aCTLs sub-population tend to decrease upon viral suppression (Figure 4). Studies have shown that despite persistently low antigenemia, high Th and E sub-population is essential to control HIV replication in the presence or absence of ART to achieve a “functional cure” [69,72]. Overall, the immunophenotype experiments proved the xfR5-D+T NP multivalent interaction results in enhanced protective immunophenotype compared to naive xfR5 mAb, D+T NPs, and xfR5 NPs; therefore, immunophenotypic differentiation indicated that mAb+ARV NP could also overcome the CM ⇒ TM ⇒ EM rate-limiting steps to promote high E and aCTL population maintenance during HIV; however, a more detailed study in an animal model is needed to confirm these preliminary results.

Our proof-of-concept study demonstrates that the novel CCR5 mAb decorated ARV-loaded nanoformulation (xfR5-D+T NPs) could potentially be a single delivery system providing two-layered protection. The competitive binding to CCR5 on target cells provides the first layer of protection by blocking HIV entry (the primary target of HIV to invade the host cell), and the next level of protection comes from eventual ART-mediated blockage of HIV replication and integration. Our preliminary study at the immunological level indicates that prolonged CCR5 receptor blockage and ART protection may also contribute to inducing a protective anti-HIV immune response, i.e., stimulation of protective immunophenotype that could prevent and promote immune cell induce killing of fresh or latent HIV-infected cells. A dual protective mechanism may prevent HIV infection from spreading from the naive and latent CCR5+ cell population. Thus, this could be a potential strategy for achieving an “HIV-functional cure.”

## 5. Conclusions

We formulated a dual-action targeted nanoformulation, i.e., xfR5-D+T NP. The ex vivo system (primary hPBMCs) experiments demonstrated xfR5-D+T NP induced CCR5 blocking and intracellular ARV drug release, which provides two levels of protection against new HIV infection or latently infected HIV+ cells. Further, xfR5-D+T NP treatment also promotes reprogramming of the immune repertoire function of memory T-cells and could help reconstitute anti-HIV immunity. Based on the proof-of-concept study, we conclude that xfR5-D+T NP combines the advantages of ART and HIV entry blockage, which augments anti-HIV immune reconstitution. Thus, the dual-action strategy could be a promising multifactorial strategy to target the HIV latent reservoir to achieve a “functional cure.”

## Figures and Tables

**Figure 1 nanomaterials-12-01942-f001:**
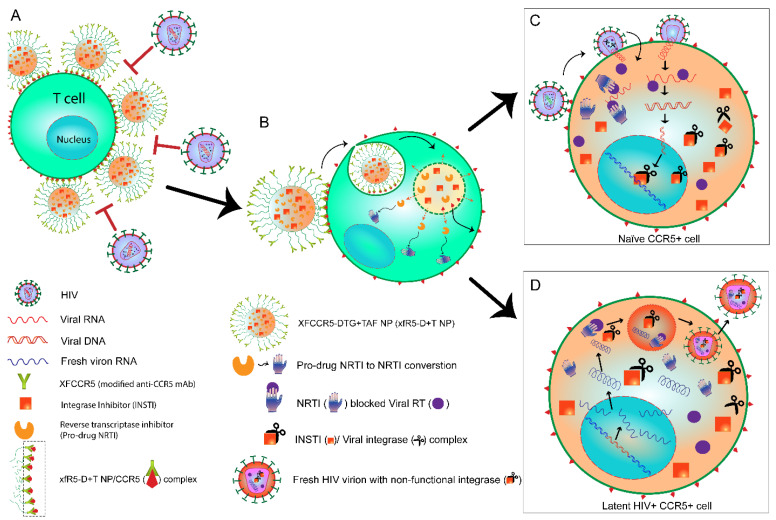
Schematic diagram explaining the dual-action strategy of targeted cARV-nanoformulation. (**A**) The first level of protection: Anti-CCR5 mAb+ ARVs loaded NP bound on the T-cell surface via CCR5 receptor (xfR5-D+T NP/CCR5 complex) blocks HIV interaction with CCR5 on the T-cell surface by two approaches, i.e., first, by blocking HIV from CCR5 binding and the second, providing steric hindrance to HIV virions due to nanoformulation crowding on T-cell surface (red block arrows) [30]. (**B**) The xfR5 D+T NP/CCR5 complex triggers the CCR5 trafficking pathway due to anti-CCR5 mAb binding with the CCR5+ T-cells [31], leading to the internalization of xfR5 D+T NP. Within the endosome, the low pH induces the degradation of polymeric NPs, leading to the release of DTG (INSTI)and TAF (NRTI) into the cytoplasm. The second level of protection: the presence of an intercellular pool of TFV di-phosphates and DTG to prevent (**C**) naive cell HIV infection. (**D**) The latently HIV+ cells with free intracellular high-affinity DTG (INSTI), will bind with fresh integrase enzymes produced during the reactivation stage, resulting in a non-functional INSTI/integrase complex; therefore, fresh HIV virions thus produced will be with non-functional INSTIs, and, fresh virion will not be able to integrate viral DNA into the host genome.

**Figure 2 nanomaterials-12-01942-f002:**
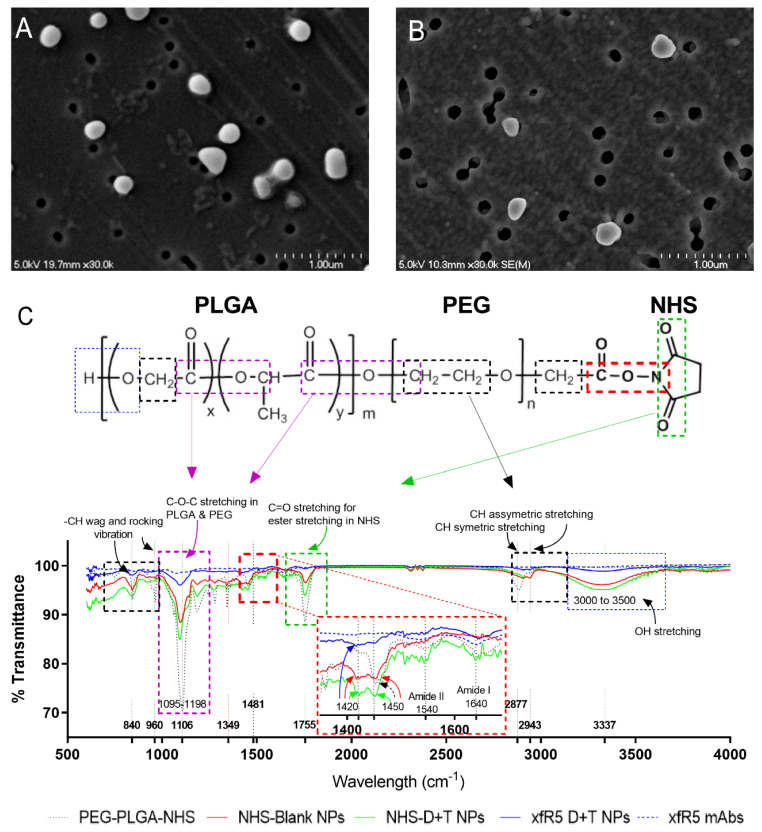
Representative scanning electron microscopic image of NHS-D+T NP (**A**) and xfR5 D+T NP (**B**). (**C**) Representative FT-IR spectrum obtained from PEG-PLGA-NHS polymer, NHS functionalized NPs (NHS-Blank NPs), NHS functionalized D+T NPs (NHS-D+T NPs), xfR5 D+T NPs, and xfR5 mAbs. The significant functional group bands are presented in various color boxes (Orange: –CH wag and rocking vibration band (840–900 cm^−1^); Purple: C-O-C stretching bands (1095–1198 cm^−1^); Red: amide bond; Green: C=O stretching band (1695–1818 cm^−1^); Red: –CON– stretching based on reported FT-IR spectral studies [46,47]). Inset graph enlargement image (**C**) shows NH–band shifting from 1450 cm^−1^ to 1420 cm^−1^ indicating COO–NC bond conversion to amide bond (–CONH–); and the presence of Amide I and II at 1540 cm^−1^ and 1640 cm^−1^, respectively.

**Figure 3 nanomaterials-12-01942-f003:**
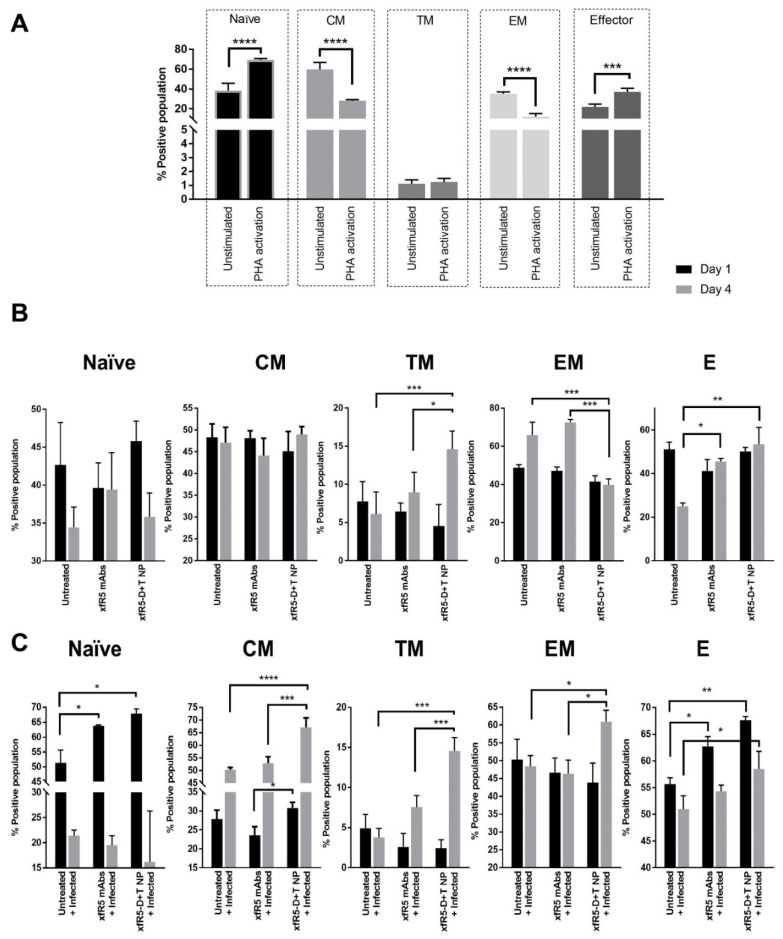
T-cell differentiation phenotype under (**A**) untreated, (**B**) protection against HIV challenge (HIV prophylaxis), and (**C**), HIV infected T-cell treatment (HIV treatment) condition. (**A**) The immunophenotypic differentiation (after day 1 and day 4) of untreated-nonactivated PBMCs was compared with PHA-activated PBMCs. (**B**) Immunophenotyping pattern evaluation after cell treatment followed by HIV challenge condition. The immunophenotypic differentiation of untreated PBMCs was compared to xfR5 mAbs, and xfR5 D+T NPs were compared on day 1 (after 1 day of treatment) vs. day 4 (after HIV-1_ADA_ virus challenge). (**C**) Immunophenotyping pattern evaluation of HIV infected followed by treatment condition. The immunophenotypic differentiation of untreated+infected PBMCs was compared to infected PBMCs treated with xfR5 mAbs, or xfR5 D+T NPs were compared on day 1 vs. 4, after 1 day of treatment. The differentiation pattern was evaluated following naïve, CM, TM, EM, and E sub-population as depicted in the x-axis, whereas the y-axis represents respective marker% positive cells. The data are presented as mean ± SEM of three independent experiments on three healthy donors (*n* = 3). The significance was represented by the asterisk (*) symbol, where ‘*’, ‘**’, ‘***’ and ‘****’ correspond to *p* values <0.05, <0.01, <0.001 and <0.0001, respectively.

**Figure 4 nanomaterials-12-01942-f004:**
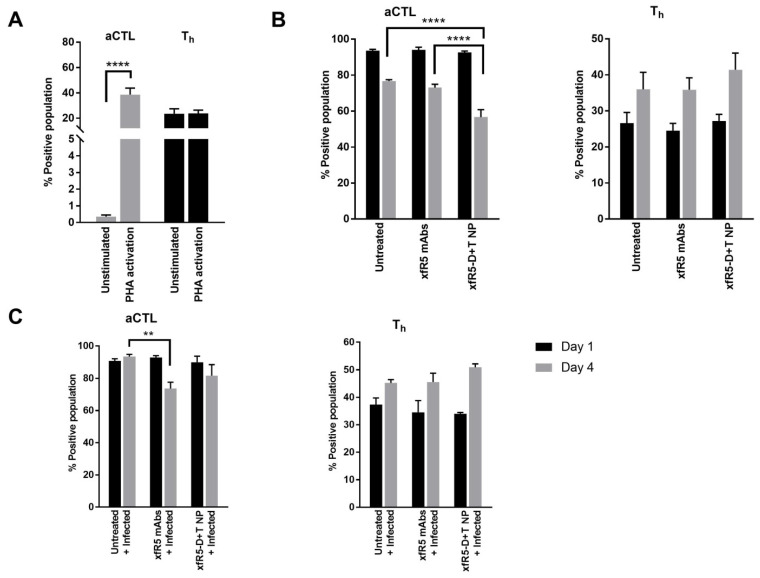
Effects on effector (E T-cell) phenotype upon treatment (with xfR5-D+T NP and xfR5 mAbs) compared to untreated conditions during PrEP and HIV-1 infection. (**A**) Comparative effect on aCTL (CD8+ E T-cell) and T_h_ (CD4+ E T-cell) phenotype under unstimulated and stimulated (PHA-activated). (**B**) Under HIV challenge condition: aCTL (left graph) and T_h_ (right graph) population on day 1 (black bar) and day 4 (gray bar). (**C**) The treatment effect in the presence of HIV-1 infection, on aCTL (left graph) and T_h_ (right graph) on day 1, and day 4. Each data set represents the mean ± SEM of three independent experiments on PBMCs obtained from 3 healthy donors (*n* = 3). The gating strategy for this study has been explained in Appendix A. The significance is represented as the asterisk (*) symbol, where ‘**’ and ‘****’ correspond to *p* values <0.01 and <0.0001, respectively.

**Table 1 nanomaterials-12-01942-t001:** Detailed information about different T lymphocyte phenotypic markers used for the immunophenotypic study.

T-Lymphocyte Target	Marker	Fluorescence Dye Conjugated	Monoclonal Antibody Type	Company
T lymphocytes	CD3	Alexa Flour 594	Mouse anti-human	BioLegend
Helper T-cells	CD4	PerCP-Cy5.5	Mouse anti-human	TONBO biosciences
		Alexa Flour 700	Mouse anti-human	BioLegend
Cytotoxic T-cells	CD8	PE/Cy7	Mouse anti-human	BioLegend
Monocytes	CD68	APC	Mouse anti-human	Invitrogen
HIV target on T-cell	xfR5	Cy3	Modified anti-human	ATCC
Transition T-cells	CCR7	CCR7-APC eFlour 780	Mouse anti-human	Invitrogen
Memory T-cells	CD45RO	Pacific Blue	Mouse anti-human	BioLegend
Activated T-cells	CD69	Alexa Flour 488	Mouse anti-human	BioLegend
HIV maker on T-cells	CCR5	APC	Mouse anti-human	BioLegend
Intermediate memory T-cells	CD27	Alexa Flour 700	Mouse anti-human	BioLegend
DC or latently infected T-cells	CD2	Pacific Blue	Mouse anti-human	BioLegend

**Table 2 nanomaterials-12-01942-t002:** Physicochemical characteristics of xfR5-D+T NPs.

Type	Size (nm)	Surface Charge (mV)	Polydispersity Index (PDI)	%Drug Entrapment Efficiency (%*EE*)	xfR5 mAb Bound per mg D+T NP
NHS-D+T NP	198.7 ± 10	−28.15 ± 1.9	0.15 ± 0.01	DTG: 58.8 ± 10 TAF: 60.5 ± 10.3	N/A
xfR5-D+T NP	212.6 ± 20.7	−17.77 ± 1.9	0.20 ± 0.018	DTG: 58.5 ± 9 TAF: 60.4 ± 10.6	3.7 ± 0.52 µg

Data represented as mean ± SEM, obtained from three independent batches of NHS-D+T NPs or xfR5-D+T NPs (*n* = 3); N/A = not applicable.

**Table 3 nanomaterials-12-01942-t003:** Comparative binding affinity (K_d_) of xfR5-D+T NPs vs. xfR5 mAb with TZM-bl cells (CD4+ CCR5+ cell line), primary CD4+ T-cells (T_h_ cells), and with other HIV-1 prone and CCR5 receptor-expressing immune cells (primary CD8+, CD68+, and CD2+ T-cells).

Type	Binding Affinity (K_d_, nM)				
	TZM-bl Cells	CD4+ T-Cells	CD8+ T-Cells	CD68+ T-Cells	CD2+ T-Cells
xfR5-D+T NPs	0.038 ± 0.020	4.31 ± 1.47	0.99 ± 0.38	0.11 ± 0.066	1.5 ± 1.1
xfR5 mAbs	0.25 ± 0.15	20.87 ± 10.65	25.02 ± 14.3	0.21 ± 0.10	2.6 ± 2.5
Wild-type anti-CCR5 mAb	2.02 ± 0.66	ND	ND	ND	ND

Data represented as mean ± SEM obtained from three healthy independent donors; ND = not determined.

**Table 4 nanomaterials-12-01942-t004:** Comparative intracellular kinetics study of DTG, TAF, active drug (TFV), and its metabolite (TFV-dp) upon xfR5-D+T NPs or D+T solution treatment.

Parameter	Units	NP				Solution			
		TAF	TFV	DTG	TFV-dp	TAF	TFV	DTG	TFV-dp
C_max_	(pmole/10^6^ cells)	18.1 ± 3.3	1250.2 ± 269.1	37,027.8 ± 5401.0	29.7 ±13.1	1.4 ± 0.7	1428.4 ± 580.4	334.0 ± 197.4	12.9 ± 4.1
AUC_all_	h*(pmole/10^6^ cells)	825.4 ± 119.5	52,384.8 ± 4613.2	2,240,490.8 ± 240,106.3	827.5 ± 125.5	11.4 ± 4.4	47,465.0 ± 9641.7	8805.0 ± 1934.1	585.9 ± 94.6
t_1/2_	h	75.5	21.6	79.2	32.3	6.5	22.3	18.0	25.4

C_max_ = maximum concentration; AUC_all_ = area under the curve; t_1/2_ = half-life of the drug dose; Data presented as mean ± SEM of three independent experiments (*n* = 3), with each experiment with three repeats/variable.

**Table 5 nanomaterials-12-01942-t005:** Comparative cytotoxicity (*CC*_50_), HIV inhibition (*IC*_50_), and selectivity index (SI, *CC*_50_/*IC*_50_) study of xfR5-D+T NPs vs. xfR5 mAb on TZM-bl cells and primary CD4+ T-cells.

Cell Type	Treatment Type	*CC*_50_ (nM)	*IC*_50_ (nM)	SI
TZM-bl	xfR5-D+T NPs	2910 ± 134.9	0.035 ± 0.01	82,670
	xfR5 NPs	998.7 ± 122.2	0.055 ± 0.01	18,254
	D+T NPs	3095 ± 102.3	0.2 ± 0.16	15,872
	xfR5 mAb	1464 ± 35.5	18.53 ± 2.85	79
PBMCs	xfR5 D+T NPs	690.6 ± 114.7	9.77 ± 2.03	71
	xfR5 NPs	1428 ± 83.5	53.34 ± 2.34	26
	D+T NPs	1009 ± 88.7	33.95 ± 1.91	30
	xfR5 mAb	816.7 ± 93	115.9 ± 1.61	7

Data represented as mean ± SEM (*n* = 3, healthy independent donors); *CC*_50_ = 50% cytotoxic concentration; *IC*_50_ = half-maximal inhibitory concentration; SI: Selectivity Index.

## Data Availability

Not applicable.

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
