# Peer review of "Targeted Immuno-Antiretroviral to Promote Dual Protection against HIV: A Proof-of-Concept Study"

_nanomaterials, 2022, doi:10.3390/nano12111942_

Round 1
Reviewer 1 Report
In this work, Mandal et al. propose a proof-of-concept study based on the dual action mechanism of nanoparticles conjugated with an anti-CCR5 monoclonal antibodies and loaded with antiretroviral drugs to provide two levels of protection against new HIV infection or in latently infected HIV+ cells. The work addresses the unresolved global health problem of HIV infection and focuses on the search for a nanoformulation to achieve a "functional cure" of HIV. The rationale of the work is well explained, the experimental work is well carried out and the results and the discussion fit the proposed objectives. The manuscript is well structured, presented and is understandable to the reader. In my opinion, this work deserves to be published in the Nanomaterials Journal.
However, there are some minor suggestions and questions that should be addressed.
1- In my opinion, the last paragraph of the introduction section (lines 100-129) as well as the schematic diagram of Figure 1 could be incorporated into the discussion section since corresponds to the explanation of the proof of concept demonstration. In addition, the Figure 1 would be more understandable after reading the results and the general discussion. Regarding Figure 1, it is difficult to visualize the symbols and icons in the scheme. It would be appreciated if the images in figure 1 were larger.
2- In page 5, the sentence on lines 200-201 is not understandable.
3- I understand that the rationale for selecting the modified xfR5 mAb compared to the wild-type CCR5 mAb is its higher affinity for the co-receptor, but I would appreciate a brief description of the background of this modified mAb as well as its differential characteristics with the wild-type CCR5 mAb. Furthermore, in experimental section 2.6., it is detailed that the ratio between the Cy3 dye and the bound xfR5 mAb in the conjugated Cy3-xfR5 mAb is 3:1. This relationship should also be reported for the purchased wild-type Cy3-CCR5 mAb, as the binding affinity of both mAbs is compared in Table 3 and the dye:mAb ratio should be similar in both cases.
4- In Table 2, why is the % Drug Entrapment Efficiency (%EE) value of formulation xfR5-D+T NP not available N/A? This fundamental data should be provided in the physicochemical characterization of nanoparticles.
5- The Figure 2C, corresponding to the representative FT-IR spectra, should be larger for easier viewing.
6- The format of Table 5 should be improved since the numerical values are not read correctly.
7- On page 19, there some gaps in sentences of lines 702, 706, 721.
8- There are some incomplete references, ex. Refs. 10, 17, 22, 32, 44.
Author Response
The authors like to thank the reviewer for such a positive and encouraging response. We have included and tried our best to respond to all the corrections reviewer suggested. In the updated manuscript all corrections are highlighted in yellow. Please find the attached pdf file in response to the reviewer's comments.

Reviewer 2 Report
The article “Targeted immuno-antiretroviral to promote dual protection 2 against HIV: A proof-of-concept study” by S. Mandal and colleagues reports the preparation and characterization of ARV-loaded nanoparticles functionalized on their surface with CCR5 mAb.
The article is well-written and organized, loaded with experimental data supporting the conclusions. I whish all the research articles were of this kind.
In my opinion the article deserves publication in Nanomaterials after minor improvements and corrections.
There are many acronyms throughout the text, please check carefully that they are all explained the first time they are encountered.
Some captions in Figure 1 are too small and cannot be read properly.
Similarly, A and B in Figure 2 should be on a top row while C should be enlarged in a bottom row to make the image clearer and more readable.
Section 2.5 could be split in 3 subsections (e.g. synthesis, functionalization, and characterization) for easier reading.
Some English and editing errors are present, please correct (line 201, spaces between numbers and units, N- in chemical names goes in italics, etc.).
Rpm must be converted to either g or rcf.
Average numbers and standard deviations must have the same precision, i.e. number of decimals (e.g. lines 435, 477, tables, etc).
Author Response
The authors are thankful to the reviewer for the encouraging review response. In the updated manuscript, all corrections are highlighted in yellow. I am attaching a pdf file responding to the reviewer's comment.
